# Dbf4-Cdc7 (DDK) Inhibitor PHA-767491 Displays Potent Anti-Proliferative Effects via Crosstalk with the CDK2-RB-E2F Pathway

**DOI:** 10.3390/biomedicines10082012

**Published:** 2022-08-19

**Authors:** Tekle Pauzaite, James Tollitt, Betul Sopaci, Louise Caprani, Olivia Iwanowytsch, Urvi Thacker, John G. Hardy, Sarah L. Allinson, Nikki A. Copeland

**Affiliations:** 1Biomedical and Life Sciences, Faculty of Health and Medicine, Lancaster University, Lancaster LA1 4YQ, UK; 2Materials Science Institute, Lancaster University, Lancaster LA1 4YW, UK; 3Department of Chemistry, Faculty of Science and Technology, Lancaster University, Lancaster LA1 4YB, UK

**Keywords:** DNA replication, Dbf4-dependent kinase (DDK), chemical probes, kinase inhibition, RB^+^ cancer

## Abstract

Precise regulation of DNA replication complex assembly requires cyclin-dependent kinase (CDK) and Dbf4-dependent kinase (DDK) activities to activate the replicative helicase complex and initiate DNA replication. Chemical probes have been essential in the molecular analysis of DDK-mediated regulation of MCM2-7 activation and the initiation phase of DNA replication. Here, the inhibitory activity of two distinct DDK inhibitor chemotypes, PHA-767491 and XL-413, were assessed in cell-free and cell-based proliferation assays. PHA-767491 and XL-413 show distinct effects at the level of cellular proliferation, initiation of DNA replication and replisome activity. XL-413 and PHA-767491 both reduce DDK-specific phosphorylation of MCM2 but show differential potency in prevention of S-phase entry. DNA combing and DNA replication assays show that PHA-767491 is a potent inhibitor of the initiation phase of DNA replication but XL413 has weak activity. Importantly, PHA-767491 decreased E2F-mediated transcription of the G1/S regulators cyclin A2, cyclin E1 and cyclin E2, and this effect was independent of CDK9 inhibition. Significantly, the enhanced inhibitory profile of PHA-767491 is mediated by potent inhibition of both DDK and the CDK2-Rb-E2F transcriptional network, that provides the molecular basis for its increased anti-proliferative effects in RB^+^ cancer cell lines.

## 1. Introduction

The cell cycle is driven by the activities of cyclin-dependent kinases (CDKs) and Dbf4-dependent kinase (Dbf4-Cdc7, referred to as DDK) [1,2]. Coordination of CDK and DDK activity in G1 phase regulates loading and activation of the MCM2-7 helicase, aiding initiation of DNA replication. Dbf4-dependent kinase (DDK) is composed of Dbf4 regulatory subunit and Cdc7, a serine/threonine kinase that contributes to DNA replication initiation by phosphorylating MCM2-7 and promoting assembly of the replicative helicase CDC45, MCM2-7 and GINS (CMG) complex [3,4,5,6]. CDK- and DDK-mediated phosphorylation of Sld3 and Dpb11 are required for loading of the CMG complex [6,7,8,9] and helicase activation [10]. Subsequently, cyclin E-CDK2 and cyclin A-CDK2 activity promotes replisome formation and facilitates the initiation of DNA replication [11,12,13,14].

The use of small molecule chemical probes has provided considerable insight into the molecular mechanisms of targeted pathways [15]. However, the specificity of chemical probes needs to be fully characterised to prevent off-target effects causing misinterpretation of results [16]. The use of small molecule Cdc7 inhibitors have enabled interrogation of the role of DDK as a molecular target in cancer therapy and in the regulation of DNA replication origin activation. The Cdc7 inhibitors PHA-767491, a pyrrolopyridinone [17] and XL-413, a benzofuropyrimidinone [18], both display low nanomolar IC50 values for DDK in vitro. Atomic structures for PHA-767491 and XL-413 bound to DDK provide the molecular basis for XL413 and its improved potency and selectivity due to increased contacts between XL-413 and DDK [19]. In addition, the recent atomic structure of the DDK-MCM2-7 complex will facilitate the design of a new generation of more potent and selective DDK inhibitors [20].

The DDK inhibitors XL-413 and PHA-767491 display differential activities in the initiation phase of DNA replication. Both XL-413 and PHA-767491 modulate replication fork dynamics via different mechanisms [21,22]. PHA-767491 inhibits the initiation phase of DNA replication. In contrast, XL-413 affects fork progression without affecting the initiation phase of DNA replication [13]. In Xenopus cell-free DNA replication systems, PHA-767491 inhibits DDK-mediated helicase activation, blocking initiation of DNA replication [23]. However, PHA-767491 does not inhibit the elongation phase of DNA synthesis [17]. Importantly, the inhibition of DDK activity using a chemical genetic approach utilising an analogue-sensitive Cdc7 mutant, demonstrated that Cdc7 is an essential gene that is required for phosphorylation and activation of the MCM2-7 helicase [24].

Cdc7 is frequently overexpressed in colon, breast and lung tumours [25,26]. Targeting Cdc7 may be of clinical significance in cancer therapy. Small molecule Cdc7 inhibitors selectively promote apoptosis in cancer cells without activating apoptosis in healthy cells [26,27]. DDK inhibition is cytostatic in normal cells leading to cell cycle arrest. In contrast DDK inhibition increases apoptosis in cancer cell lines in vitro and in xenograft models [28,29,30]. Despite showing anti-tumour activity in the colorectal carcinoma cell line, Colo-205 in vitro and in xenograft models [18], XL413 inhibitory activity is limited in other cancer cell lines in contrast to PHA-767491 [31]. The DDK inhibitors PHA-767491 and XL-413 display distinct cytotoxic effects in cancer cell lines, despite similar in vitro IC50 values. Comparison of XL-413 and PHA767491 in various cancer cells lines suggests that difference in potency may be related to solubility, specificity, or bioavailability [17,28,29,31]. Yet, the precise mechanisms that underpin these differences in potency in vitro or in vivo remain to be determined.

Here, we investigate the requirements of DDK activity in the initiation phase of DNA replication using small molecule DDK inhibitors PHA-767491 and XL-413. The differential off-target effects of PHA-767491 were characterised and provide mechanistic insight that distinguishes between the activities of XL-413 and PHA-767491. In addition, this study provides mechanistic insight into the differential cytotoxic effects seen for XL-413 and PHA-767491 in vitro. These results are pertinent to their use as tools in dissection of cell cycle regulation, initiation of DNA replication and have significant implications for their use as targeted cancer therapies.

## 2. Materials and Methods

### 2.1. Tissue Culture

Mouse embryonic fibroblasts (NIH3T3) were cultured in D-MEM Glutamax II, low glucose + 1 mM pyruvate (GIBCO, Waltham, Massachusetts, USA), 10% *v*/*v* foetal calf serum and 1× penicillin, streptomycin and glutamine (GIBCO). Cells were cultured at 37 °C, 5% CO2. NIH3T3 cells were synchronised in G0 phase by serum starvation and contact inhibition for at least 48 h. NIH3T3 cells were stimulated to re-enter the cell cycle by dilution into fresh medium with a 1:4 split [14,32,33,34]. Cell cycle synchronisation was determined by EdU incorporation, protein levels by Western blotting and mRNA synthesis by RT-qPCR.

RB positive cancer cell lines: colorectal cancer cell lines SW480, SW620 and prostate carcinoma PC3 cells were cultured in D-MEM Glutamax II, high glucose, with 1 mM pyruvate (GIBCO), 10% foetal calf serum and 1% of penicillin, streptomycin and glutamine (GIBCO). Cells were cultured at 37 °C, 5% CO_2_. Cells were synchronised in M phase with 24 h incubation in 2 mM Thymidine (Sigma Aldrich, St Louis, MO, USA), cells were then released into fresh medium for 3 h and treated for a further 10 h with 100 ng/mL (0.3 µM) Nocodazole (Sigma Aldrich). Cells were subsequently washed and released from nocodazole block into fresh medium for 4 h, then treated for 8 h with PHA-767491 or XL-413, as indicated.

### 2.2. Cell-Free DNA Replication Assays

The 3T3 murine fibroblasts were synchronised in G0 using serum starvation and contact inhibition [14,32,35,36]. Replication competent nuclei and G1 extracts were harvested 17.5 and 15 h after release from G0, respectively. HeLa cells were used for S phase extracts and were synchronised using a double thymidine treatment (2.5 mM) to synchronise at early S-phase and harvested 1 h after release from the double thymidine block. For isolation of cytosolic or nuclei fractions, cells were washed with 10 mL hypotonic buffer (20 mM HEPES pH 7.8, 15 mM potassium acetate, 0.5 mM MgCl_2_, 1 mM DTT), then incubated in ice cold hypotonic buffer for 5 min. Hypotonic buffer was removed, cells were scrape-harvested and lysed using a Wheaton dounce homogeniser. Then, 7 compressions for nuclei, and 25 compressions for extracts were performed using the tight pestle. Nuclei were harvested by centrifugation at 5000× *g* for 5 min, and cytosolic extracts produced by subsequent centrifugation at 17,000× *g*. For extracts, supernatants were frozen and stored in liquid nitrogen. For nuclei, supernatants were removed, pellets were resuspended in hypotonic buffer (10 µL per plate harvested) and flash frozen in liquid nitrogen.

Cell-free DNA replication assays were performed as described [14,32,35,36]. Briefly, nuclei and extracts were incubated at 37 °C for 30 min. Nuclei were fixed in 100 µL 4% paraformaldehyde (PFA) for 15 min. Nuclei were centrifuged at 1500 RPM for 10 min through 800 µL of 30% *w*/*v* sucrose onto a polylysine-coated coverslip at the base of a scintillation tube. Coverslips were washed 3 times sequentially in PBS and 3 in times antibody buffer (10 mM PBS, 138 mM NaCl, 27 mM KCl, pH 7.4, 1% *w*/*v* BSA, 0.02% *w*/*v* SDS, 0.1% *v*/*v* Triton X-100) for 5 min. Coverslips were incubated for 30 min in a humidity chamber at 37 °C in a 1:1000 dilution of streptavidin-Alexa Fluor™ 555 Conjugate (Thermo Fisher Scientific, Waltham, Massachusetts, USA). Coverslips were washed 3 times sequentially in antibody buffer and 3 times in PBS. Coverslips were mounted in VECTASHIELD mounting medium with DAPI. Nuclei were counted using a Zeiss Scope.A1 epifluorescence microscope counting >100 nuclei per reaction. Data presented show 3 experimental repeats for each condition, showing mean ± S.D. Statistical analysis was performed in GraphPad Prism 9.3.1. The data was assessed for normality using the Shapiro–Wilk test prior to analysis using an ordinary one-way ANOVA multicomparisons using Šídák’s multiple comparisons test. **** *p* < 0.001. S vs. XL, G1 vs. PHA and XL vs. DMSO were non-significant.

### 2.3. Quantitative Real Time PCR (qRT-PCR)

Total RNA was purified using PureLink^®^ RNA MiniKit (Ambion, Austin, TX, USA) according to the manufacturer’s instructions. qRT-PCR was performed using ONE STEP qPCR kit (Invitrogen, Waltham, MA, USA) using 20 ng of total RNA and Taqman primers (Thermo Fisher Scientific, Waltham, MA, USA) for cyclin E1 (Mm00432367, Hs01026536), cyclin E2 (Mm00438077, Hs00180319), cyclin A2 (Mm00438063, Hs00996788), Dbf4 (Mm01324087), Cdc7 (Mm00438122), GAPDH (Mm03302249, Hs02758991) and 18S rRNA (Hs03003631) using a FX96 Touch™ Real-Time PCR System (Bio-Rad, Hemel Hempstead, UK). Program: cDNAsynthesis: 50.0 °C 15 min, denaturation 95.0 °C for 2 min, then 40 cycles using a 2-step program 95.0 °C for 15 s and 60.0 °C for 30 s.

### 2.4. Kinase Inhibition Assays

Asynchronous NIH3T3 cells were treated with small molecule inhibitors at 50–70% confluence at indicated concentrations (Table 1) 4 h prior to the cell harvesting. In synchronised cell cycle populations, cells were treated at 12 h after release from quiescence that is approximately 3 h before the restriction point (~15 h post release from G0 [33]) and harvested at indicated time points.

### 2.5. Western Blotting

Cells were harvested in phosphate buffered saline (Sigma Aldrich), 1 mM PMSF, 1 mM DTT, and phosphatase inhibitor cocktail Set V (Calbiochem), boiled in 4x SDS-PAGE loading buffer and resolved on 10% SDS-PAGE gels. Cells were fractionated by addition of 0.5% *v*/*v* Triton X-100 and centrifugation (5000× *g* for 5 min) and separated into cytoplasmic/nucleosolic and chromatin-associated fractions. Proteins were transferred by semi-dry transfer in 10 mM CAPS, 0.3M TRIS, 0.02% *w*/*v* SDS and 10% *v*/*v* ethanol onto nitrocellulose membrane. Membranes were blocked in TBS, 0.1% Tween (TBST) with 1% *w*/*v* BSA (TBST-BSA) and probed with specific antibodies for target proteins (Table 2) in TBST-BSA buffer. Membranes were washed 3 times in TBST-BSA and species-specific secondary antibodies were added in TBST-BSA buffer for 1 h, membranes washed 3 times in TBST and imaged with Thermo Fisher SuperSignal™ West Pico PLUS Chemiluminescent Substrate using a Bio-Rad Gel Doc XR+ imaging system.

### 2.6. Flow Cytometry

Cells were incubated with 10 µM EdU (Invitrogen) for 1 h to label nascent DNA. Cells were harvested by trypsinisation, resuspended in media and centrifuged for 5 min at 500× *g*. Cell pellets were washed 3 times with 500 µL 1% BSA in PBS followed by centrifugation for 5 min at 500× *g*, cells were fixed in 4% PFA for 15 min, washed 3x in PBS, 1% *w*/*v* BSA and permeabilised by addition of Triton X-100 to a final concentration of 0.5% *v*/*v* for 20 min. Cells were washed 2x with PBS, 1% *w*/*v* BSA prior to addition of Click-iT™ EdU Cell Proliferation Assay Cocktail (Invitrogen) with Alexa Fluor 488 Azide for 30 min. Cells were washed 2 times with 1% BSA and once with PBS, and stained with 100 µg/mL of propidium iodide in 0.1% Triton X-100 in PBS for 20 min. Flow cytometry was performed using a CytoFLEX Beckman Coulter, and compensation was applied.

### 2.7. siRNA-Mediated Depletion of Target Genes

Synchronised NIH3T3 cells were prepared by serum starvation and contact inhibition [14,32,35,36]. Synchronised cells were trypsinised, re-suspended in media and harvested by centrifugation for 5 min at 500× *g*. Media was removed and cells co-transfected with anti-Cdc7 (Mm s63747) and anti-Dbf4 (Mm s77567) (Ambion) siRNAs in Cell Line Nucleofector^®^ Solution R for NIH3T3 (Amaxa^®^ Cell Line Nucleofector^®^ Kit R; Lonza protocol, Basel, Switzerland) using program U-030. The cells were scrape harvested 20 h after the transfection for Western blot analysis and harvested by trypsinisation for RNA extraction and qPCR analysis.

### 2.8. DNA Combing

NIH 3T3 cells were grown to 60% confluence. Cells were treated with either 10 µM PHA-767491, XL-413 or 0.1% *v*/*v* DMSO for 24 h. Following the 24 h incubation, media was replaced with media supplemented with 25 µM iododeoxyuridine (IdU) for 20 min, followed by media supplemented with 250 µM chlorodeoxyuridine (CldU) for 20 min. Cells were immediately harvested by trypsinisation and centrifugation. Cells were embedded in agarose and purified using Bio-Rad CHEF Mammalian Genomic DNA Plug Kit according to the instructions. To extract genomic DNA, agarose plugs were melted into 1.5 mL MES-E solution (50 mM MES, 1 mM EDTA, pH 5.7). The DNA solution was placed into 25 × 25 × 3 mm Teflon wells. Silanised glass coverslips were generated using a procedure adapted from [36,37]. Silanised coverslips were lowered into DNA solution using a Biolin scientific KSV NIMA dip coater. Coverslips were incubated in DNA solution for 10 min. Coverslips were withdrawn from DNA solution at a constant speed of 300 µm/s. Coverslips were attached to glass slides using 4 drops of cyanoacrylate glue. Coverslips were incubated at 60 °C in a drying oven for 1 h. DNA was denatured in 0.5 M NaOH for 30 min. Each wash step consisted of 3 × 3 min washes in 0.2 µm filtered PBS-T. Coverslips were blocked in 3% BSA PBS-T at 37 °C for 1 h and incubated overnight in a humidity chamber at 4 °C in 50 µL of primary antibody solution (anti-CldU, anti-IdU in blocking buffer). Coverslips were washed, incubated for 30 min at 37 °C in 3% BSA TBST with AlexaFluor 488 and AlexaFluor 633 secondary antibodies (Table 3). Coverslips were washed, incubated at room temperature for 2 h in 3% BSA, TBST with anti-ssDNA. Coverslips were washed then incubated for 30 min at 37 °C in the Alexafluor 568 secondary antibody solution. Coverslips were washed. Coverslips were mounted using prolong gold mounting media.

DNA was imaged using a Zeiss 880 confocal microscope using the 63x objective. Images were taken in three tracks ssDNA, IdU and CldU excited using the 568, 633 and 488 nm lasers, respectively, and captured using 568–640, 638–747 and 493–598 nm ranges. Replication tracks were measured using the Fiji software package. In total, >150 replication tracks were measured per sample. Significance was determined using an unpaired 2 tailed *t*-test (GraphPad, San Diego, CA, USA).

### 2.9. PrestoBlue Cell Proliferation Assays

PC3, SW480 and SW620 cells were dissociated using trypsin-EDTA, collected in an equal volume of media, harvested by centrifugation at 500 rpm for 5 min. Cells were resuspended in 1 mL of media and cell counts performed using 10 µL of the cell suspension using a Neubauer improved haemocytometer. Cell number optimisation was performed by starting with a cell stock 200,000 cells/mL and serially diluting cells across a 96-well plate. In total, 2000 cells per well was selected, cells were plated onto a 96-well fluorescence plate and cultured for 24 h prior to addition of the kinase inhibitors, as described. After addition of kinase inhibitors, 96-well plates were incubated for 48 h at 37 °C, 5% CO_2_ and 10 µL PrestoBlue^TM^ Cell Viability Reagent (Invitrogen) was added to 90 µL of cells/media. Plates were incubated at 37 °C, 5% CO_2_ for 1 h. Fluorescence excitation occurred at 560 nm and emissions were collected at 598 nm using a 96-well plate reader (Tecan Infinite^®^ M200 PRO, Männedorf Switzerland). Background absorbance was corrected and fluorescence signal was plotted against concentration of the inhibitor using GraphPad Prism. All IC50 values were determined using nonlinear regression and a dose–response inhibition plot was created (3 parameter) using fluorescence levels of 9 independent experiments showing mean ± standard deviation.

## 3. Results

### 3.1. PHA-767491 Is a Potent Inhibitor of the Initiation Phase of DNA Replication

To evaluate the effect of the DDK inhibitors XL-413 and PHA-767491 on cell cycle dynamics, flow cytometry of 3T3 cells was performed after drug treatment. Each inhibitor was added for 24 h prior to thymidine analogue EdU labelling (Figure 1A). PHA-767491 reduced the number of cells in S-phase and the fluorescence intensity of initiated cells (2.7% vs. 40.7%) (Figure 1B). In contrast, XL-413 increased the number of cells in S-phase (57%) (Figure 1A). Comparison of PHA-767491 and solvent controls showed a reduction in the S-phase population and an increase in G2 population, whereas the effect of XL-413 was less pronounced (Figure 1A,B). Similarly, using a post-quiescent synchronised population of 3T3 cells treated with PHA-767491 and XL-413 for 12 h post release from G0 revealed a differential effect for PHA-767491 and XL-413. Control cells showed ~37% cells in S-phase, whereas PHA-767491 efficiently reduced the number of cells in S-phase (8%) and XL-413 showed an intermediate effect (24%, Figure 1C). Both XL-413 and PHA-767491 significantly reduced S-phase cells relative to control from 18 h after release (*p* < 0.001).

To establish the effect of each DDK inhibitor on the initiation of DNA replication, a cell-free mammalian DNA replication system was used that recapitulates the initiation phase of DNA replication [14,32,33,35,36]. G1 cytosolic extracts and synchronised late G1 nuclei were used as a control (Figure 1D) and identified the proportion of cells that were in S-phase, i.e., in the elongation phase of DNA synthesis. To identify the initiation competent population of cells, S-phase cytosolic extracts were added to licensed late G1 nuclei and ~47% of nuclei entered S-phase (biotin—dUTP incorporation; Figure 1D). Addition of DDK inhibitors PHA-767491 and XL-413 to in vitro DNA replication assays revealed that PHA-767491, but not XL-413, reduced the number of nuclei that initiated DNA replication (Figure 1D). PHA-767491 reduced the number of initiating nuclei to ~12% consistent with the G1 extract control, whereas XL-413 did not significantly affect initiation of DNA replication, relative to positive control (bar 2) and solvent control (bar 5) reactions.

To gain further insight into the replication dynamics and the effect of each DDK inhibitor, DNA combing was performed. Initially, the replication event density was determined using EdU to label nascent DNA replication (Figure 1E,F). Significantly, PHA-767491 reduced DNA replication to levels that were not detected using DNA combing. However, XL413 reduced the frequency of origin usage to ~50% of control (Figure 1F) and significantly reduced DNA replication fork rate from 1.39 ± 0.89 to 1.08 ± 0.0.55 kbp/min (*p* = 0.002; Figure 1 G,H). The reduction in fork rate and origin usage are consistent with reduced bulk DNA synthesis seen by flow cytometry. Taken together, both cell-free and cell-based approaches showed that PHA-767491 is a potent inhibitor of the initiation phase of DNA replication, whereas XL-413 only had a moderate effect.

### 3.2. PHA-767491 Inhibits E2F-Regulated Transcription of Cyclin E1, Cyclin E2, Dbf4 and Cyclin A2

Next, the effect of targeted inhibition of DDK and the effect on cell cycle regulators was determined. Synchronised G1 cells were treated from 12 to 20 h after release from G0 with PHA-767491 and XL-413 (Figure 2A). DDK phosphorylates MCM2 at Ser40 and Ser53 [5,17,18]. Both DDK inhibitors significantly reduced MCM2 phosphorylation at pSer53-MCM2 and the CDK2/DDK site ser40/41 (lanes 2 and 3) consistent with efficient DDK inhibition [5,17,18]. However, PHA-767491 efficiently reduced RB phosphorylation on Ser811, consistent with a reduction in CDK2 activity (Figure 2A). Dbf4 and cyclin E levels were unaffected by addition of either DDK inhibitor; however, unexpectedly, PHA-767491 reduced the cyclin A protein levels relative to control and XL-413-treated cells (Figure 2A). As the RB-E2F pathway regulates cyclin A expression, this suggested that PHA-767491 may be acting via off-target mechanisms that are affecting transcription of specific genes including cyclin A. As PHA-767491 can inhibit DDK, CDK9 and CDK2 [17], it is feasible that this reduction in cyclin A levels may be related to inhibition of RNA polymerase II [38] or via inhibition of the Rb-E2F pathway [39]. Next, to evaluate the potential of PHA-767491 in reducing CDK2 activity, the DDK inhibitors PHA-767491 and XL-413 were compared to CDK2 inhibitors roscovitine and CVT313 (Figure 2C). Here, PHA-767491 shows dual functionality, reducing DDK-mediated phosphorylation of MCM2 and at CDK sites on RB (Figure 2C). Both PHA-767491 and CDK2 inhibitors potently reduce cyclin A levels relative to control and XL413, suggesting that the differential activity seen for PHA-767491 and XL-413 is mediated by off-target CDK2 inhibition by PHA-767491.

The specific reduction in cyclin A protein levels after PHA-767491 and CDK2 inhibition suggested that PHA-767491 may reduce cyclin A2 transcript levels via inhibition of the Rb-E2F pathway. To determine the mechanistic basis for this effect, E2F-regulated transcripts cyclin A2, cyclin E1, cyclin E2 and Dbf4 were monitored in a synchronous post-quiescent population of 3T3 cells (Figure 2D). This showed a synchronous increase in E2F-regulated transcripts cyclins E1, E2, A2 and Dbf4 as cells approach the G1/S transition (Figure 2D). Next, to assess whether PHA-767491 inhibits E2F transcription, 3T3 cells were treated with XL-413 and PHA676491 from 12 to 20 h post-G0 (Figure 2E). This revealed that PHA-767491, but not XL-413, significantly reduced cyclins E1, E2, A2 and Dbf4 mRNA levels (*p* < 0.001). This observation is likely related to off-target activity against CDK2 [17] and is unrelated to its DDK inhibitory activity. Consequently, to determine whether Dbf4-Cdc7 inhibition could affect cyclin A2 transcription, siRNA targeting both Cdc7 and Dbf4 were co-transfected into synchronised NIH3T3 cells and transcript levels were determined at 20 h after transfection. In this context, Cdc7 and Dbf4 transcript levels were reduced by 50 and 80%, respectively (Figure 2F). In contrast, siRNA-mediated depletion of Cdc7 and Dbf4 does not significantly affect cyclins A2, E1 or E2 expression (Figure 2G). Cdc7-Dbf4 depletion reduced DDK-mediated phosphorylation in MCM2 (pSer53 and pSer40/41), consistent with a reduction in DDK activity (Figure 2H). Additionally, depletion of Dbf4 and Cdc7 depletion did not decrease RB pS811 levels (Figure 2H), cyclin A2 transcript levels (Figure 2D) or affect cyclin A protein levels in cytoplasm or chromatin fraction (Figure 2H). The data presented show that inhibition of DDK activity by either siRNA-mediated depletion or the small molecule DDK inhibitor XL-413 do not affect cyclin E or cyclin A transcription or protein levels. These results suggest that PHA-767491 affects cyclin E and cyclin A expression independently from its DDK inhibitor activity, consistent with an off-target CDK2 inhibitory activity that could contribute to its increased activity relative to XL-413.

### 3.3. PHA-767491 Reduces E2F-Regulated Transcripts with Greater Potency Than CDK2 Inhibitors, Suggesting That Crosstalk with CDK2 Affects the RB-E2F Pathway

PHA-767491 is a potent inhibitor of CDK9 [17] that regulates RNA polymerase II activity. To assess whether PHA-767491 affects transcript levels due to global inhibition of transcription as a result of CDK9 inhibition, or via CDK2-mediated regulation via the RB-E2F pathway, the level of cyclins E1, E2 and A2 mRNA was determined relative to two housekeeping transcripts that are transcribed by RNA polymerase I (18s rRNA) and RNA polymerase II (GAPDH). If RNA pol II transcription is mediating the observed reduction in transcript levels, standardisation to an RNA polymerase II regulated transcript (GAPDH) would show differential relative quantitation when compared to 18s rRNA that is transcribed by RNA polymerase I. In synchronous cells treated with PHA-767491 from 12 to 20 h after release from G0, there was a significant decrease in cyclins A2, E1, E2 and Dbf4 transcripts relative to either GAPDH or 18S rRNA (*p* < 0.001 for cyclins A2, E1 and E2) (Figure 3A), whereas there were no significant decreases in E2F-regulated transcripts for XL-413-treated cells (Figure 3B). Importantly, comparison of relative quantitation (RQ) of transcript levels for GAPDH and 18S rRNA were found to be non-significant (*p* > 0.8 for each transcript standardised for 18S vs. GAPDH). The data suggest that PHA-767491 inhibits E2F target gene transcription via the RB-E2F pathway.

The data presented in Figure 3 suggest that PHA-767491 may affect transcription of cyclin A2 via the inhibition of CDK activity and suppression of the E2F pathway. To determine whether PHA-767491 has similar potency to CDK2 inhibitors (Roscovitine and CVT-313) in reducing E2F-mediated transcription, three E2F1-3-regulated transcripts were monitored. Each inhibitor was titrated at 3 concentrations pre-restriction point (12–20 h) after release from quiescence (G0) and cyclin E1, cyclin E2 and cyclin A2 transcript abundance was determined. PHA-767491 efficiently reduced cyclin expression at concentrations ≥ 5 µM, with less than 20% transcript levels detected (*p* < 0.001 for 5 and 10 µM; Figure 4A). This contrasted with XL-413, where the transcript levels were unaffected at all concentrations tested (Figure 4B). The use of CDK2 inhibitors CVT-313 and Roscovitine were able to reduce cyclin expression to less than 20% at 10 µM (CVT) and 30 µM (Roscovtine), respectively (Figure 4C,D). There is a concentration-dependent inhibition of E2F-mediated transcription of cyclin E and cyclin A by PHA-767491, CVT-313 and Roscovitine. In contrast, XL-413 did not affect E2F transcript abundance (Figure 4B,F). These data demonstrate that PHA-767491 (IC50 CDK2, 240 nM) is a more potent inhibitor of E2F-mediated transcription than the bona fide CDK2 inhibitors roscovitine (IC50 CDK2, 700 nM) and CVT313 (IC50 CDK2, 500 nM), consistent with their IC50 values. Furthermore, these data suggest that the differential potency between XL-413 and PHA-767491 in cancer models [31] may be related to the observed additional inhibitory crosstalk between DDK and CDK networks mediated by PHA-767491.

### 3.4. PHA-767491 Reduces E2F-Mediated Transcription of Cyclins E and A in RB-Positive Cancer Cell Lines

PHA-767491 is a potent inhibitor of cancer cell proliferation in xenograft models and in vitro via inhibition of DDK activity [17,31]. To establish if PHA-767491 could also inhibit CDK2 and reduce E2F1-3-mediated transcription in three RB^+^ cancer cell lines PC3 [40], SW480 [41] and SW620 [42] cells were treated with either CDK inhibitors (CVT313 and roscovitine) or DDK inhibitors (XL-413 and PHA-767491) (Figure 5A–C).

In PC3 cells, PHA-767491 (*p* < 0.001 cyclins E1, E2 and A2) or CDK2 inhibitors roscovitine and CVT313 reduced transcript levels relative to controls by 10–50% (Figure 5A). The addition of XL-413, however, did not significantly affect transcript levels for cyclins E1, E2 or A2 (Figure 5A–C). In SW480 stage 2 colorectal cell lines, PHA-767491 was a more potent inhibitor of E2F pathway transcripts (*p* values for cyclins E1, E2 and A2 < 0.05) when compared with XL-413, showing equivalent activity to roscovitine (*p* < 0.05) and CVT313 (*p* < 0.05). Similarly, in SW620, a metastatic colorectal carcinoma cell line, XL-413 did not significantly reduce E2F pathway transcripts, whereas PHA-767491 and CDK2 inhibitors reduced cyclin and A2, E1 and E2 transcript levels (Figure 5C). The data demonstrates that PHA-767491 has potent activity in reducing transcript levels of the G1/S cyclins A2, E1 and E2, consistent with its anti-proliferative effects in cancer models.

Next, the effect of DDK and CDK inhibitors on protein levels was determined. DDK inhibitors PHA-767491 and XL-413 efficiently reduced the DDK-mediated phosphorylation of pS53 MCM2 consistent with the efficient inhibition of DDK activity in PC3, SW480 and SW620 cell lines (Figure 5D–F). Phosphorylation of MCM2 at ser40/41 that report on both DDK and CDK activity were reduced in all contexts (Figure 4D–F). The protein levels for Dbf4 and cyclin E were unaffected by 8 h of DDK or CDK inhibitor treatment that reduced cyclin A levels in all treatments except XL-413 (Figure 5D–F). Both Western blot and qPCR show differential PHA-767491 and XL-413 activities. PHA-767491 reduces cyclin A levels at both the mRNA and the protein levels. In contrast, XL-413 does not reduce cyclin A mRNA or protein levels. These observations establish that PHA-767491 effectively and efficiently reduces DDK and CDK2 activity.

### 3.5. PHA-767491 Is a More Potent Inhibitor of Cell Cycle Progression Than XL-413 in RB-Positive Cancer Cell Lines

PHA-767491 is a more potent inhibitor of proliferation than XL-413 in cellular proliferation assays (Figure 1A,B), in vitro DNA replication assays (Figure 1C), and has higher efficacy in a range of tumour types [31]. The anti-proliferative effect of PHA-767491 and XL-413 was assessed for PC3, SW480 and SW620 cell lines. As PHA-767491 has off-target inhibitory activity against CDK2, CDK2 inhibitors were used for comparison. In each case, solvent controls were used for each cell line and analysed by dual labelling cells for total DNA (PI) and nascent DNA replication (EdU) to determine cellular proliferation (Figure 6A). PHA-767491, Roscovitine and CVT-313 reduced cell cycle progression in all cell lines, with a <75% reduction in S-phase cells in each cell line relative to controls (Figure 6D–F). However, XL-413 reduced fluorescence intensity but did not reduce the number of S-phase cells and in some cases increased the proportion in S-phase (Figure 6A–C). Significantly, PHA-767491 reduced the number of cells in S-phase in all cell lines. This is consistent with PHA-767491 targeting CDK2 and DDK activities and suggests that PHA-767491 has increased potency due to its increased repertoire of inhibitory activities. To assess the differential cytotoxic effects of DDK inhibition via XL-413 and PHA-767491, IC50 values for each inhibitor were determined. These revealed an approximate 50–100-fold increase in potency for PHA-767491 relative to XL-413 (Figure 6G–I and Table 4). It is notable that the IC50 for PHA-767491 shows similar activity to the observed inhibition of the E2F-mediated transcription of key cell cycle regulators, suggesting that the differential inhibitory effects are mediated by additional repertoire of off-target kinases for PHA-767491 in addition to DDK inhibition.

## 4. Discussion

The use of small molecule inhibitors to dissect biological processes provides powerful tools to understand cellular activity. However, the full characterisation of small molecules is required to ensure that the effects of specific inhibitors can be attributed to on-target activity. The selection of small molecules should include using different chemotypes and ensuring reproducible activities with siRNA and chemical biology approaches [15,43]. Here, differential effects were characterised using 2 DDK inhibitors that display similar nanomolar IC50 values (10 nM PHA-767491, 3.4 nM XL-413) in vitro [24,26] and have potent activities in cell-based assays in the low micromolar range (on average 3.17 μM PHA-767491 in 61 human cell lines, 2.7 μM in the Colo-205 cell line) [15,43]. However, despite the similar IC50 values for DDK inhibition in vitro and in vivo, several studies have found that PHA-767491 is more potent against cancer cell lines with respect to XL-413 activity [19,28]. This research showed that PHA-767491 has greatly increased potency against RB-positive cancers than XL with IC50 values ranging around 1 μM for PHA767491 and 92–128 μM for XL413.

Here we show that, consistent with other work, PHA-767491 and XL-413 have a similar activity when monitoring phosphorylation of MCM2 in cell-based assays suggesting that phosphorylation of MCM2 at S53 or S40/41 is a poor indicator of efficacy. However, PHA-767491 significantly reduces transcription of cyclin A, cyclin E and Dbf4 and reduces phosphorylation of Rb, with similar efficacy to small molecule CDK2 inhibitors (Figure 4). The data presented here clearly demonstrate that PHA-767491 has potent off-target activity against CDK2. The inhibition of CDK2 was detected by inhibition of RB phosphorylation and the subsequent inhibition of Rb-E2F-regulated transcription (Figure 3 and Figure 4). This correlates with inhibition of E2F transcription and a reduction in cell cycle progression as determined by flow cytometry (Figure 1 and Figure 5A–C) and IC50 values in the low micromolar range (Table 4). XL-413 does not significantly affect E2F transcription and shows limited reductions in cell cycle progression, beyond reducing the rate of EdU incorporation, which leads to a reduced signal intensity (Figure 1A and Figure 5). XL-413 reduces DNA replication origin firing and reduces replication fork speed [24]; therefore, our data and others are consistent with an increase in S-phase length (Figure 6 D–F). The data suggest that the observed potency of PHA-767491 is due to inhibition of both DDK and CDK activity that potently inhibits the initiation phase of DNA replication (Figure 1D). This inhibitory effect on CDK2 activity is well documented but the molecular basis for this activity was previously not fully described. The IC50 value for CDK2 activity for PHA-767491 of 200 nM [17], compares with 600 and 500 nM for roscovitine and CVT-313, respectively, and PHA-767491 shows this effect on RB-positive cancer cells lines PC3, SW480 and SW620 (Figure 5 and Figure 6). The data presented here shows that PHA767491 inhibits both DDK and CDK2 activities. This dual inhibitory effect acts on transcriptional regulation of the E2F pathway as well as targeting helicase activation and initiation of DNA replication. These data suggest that these activities underpin PHA-767491 as a potent inhibitor of tumour proliferation due to its broad-spectrum activities (Figure 7).

On a cautionary note, our data suggest that the use of PHA-767491 as a molecular probe of DDK function in the G1/S transition should be interpreted with caution due to its complex inhibitory profile. The use of PHA-767491 requires additional controls to ensure interpretations are robust. In particular, the effects on E2F-regulated transcripts should be monitored to elucidate any effects via CDK2 inhibition. In Xenopus systems that utilise maternally derived transcripts prior to the mid-blastula transition [44,45], PHA-767491 will not affect cyclin expression via the E2F pathway and consequently should not affect cyclin transcript abundance. However, due to dual inhibition of DDK and CDK2, caution should be used when analysing defects in the initiation phase of DNA replication where CDK2 activity is crucial for cell cycle progression.

The data presented here identified specific off-target effects for PHA-767491 that provides mechanistic insight that distinguishes the differential effects of PHA-767491 and XL-413. The observed inhibition of the E2F pathway suggests that PHA-767491 has potential for increased potency in RB+ tumours. The potent inhibition of CDK2 activity at 5 µM PHA-767491, reduces E2F-mediated transcription by >90% (Figure 6D). In contrast, DDK inhibition with XL-413 does not affect RB phosphorylation or E2F-mediated transcription, that promotes an increase in S-phase cells and markedly reduces DNA replication fork length and density.

Furthermore, our data shows that PHA-767491 inhibits the CDK2-RB1-E2F pathway with the similar efficiency to widely used CDK2 inhibitors roscovitine and CVT-313 and not via inhibition of CDK9 and RNA Polymerase II (Figure 4) [23,24]. However, this work emphasises the need for thorough and comprehensive analysis of possible off-targets of small molecule inhibitors in cell-free and cell-based conditions, as well as the potential side effects of the inhibitors. Finally, the potent anti-cancer effects in the low micromolar range for PHA-767491 are mediated via potent DDK and CDK2 inhibitory activity that requires further investigation to determine its clinical significance in RB^+^ tumours.

## Figures and Tables

**Figure 1 biomedicines-10-02012-f001:**
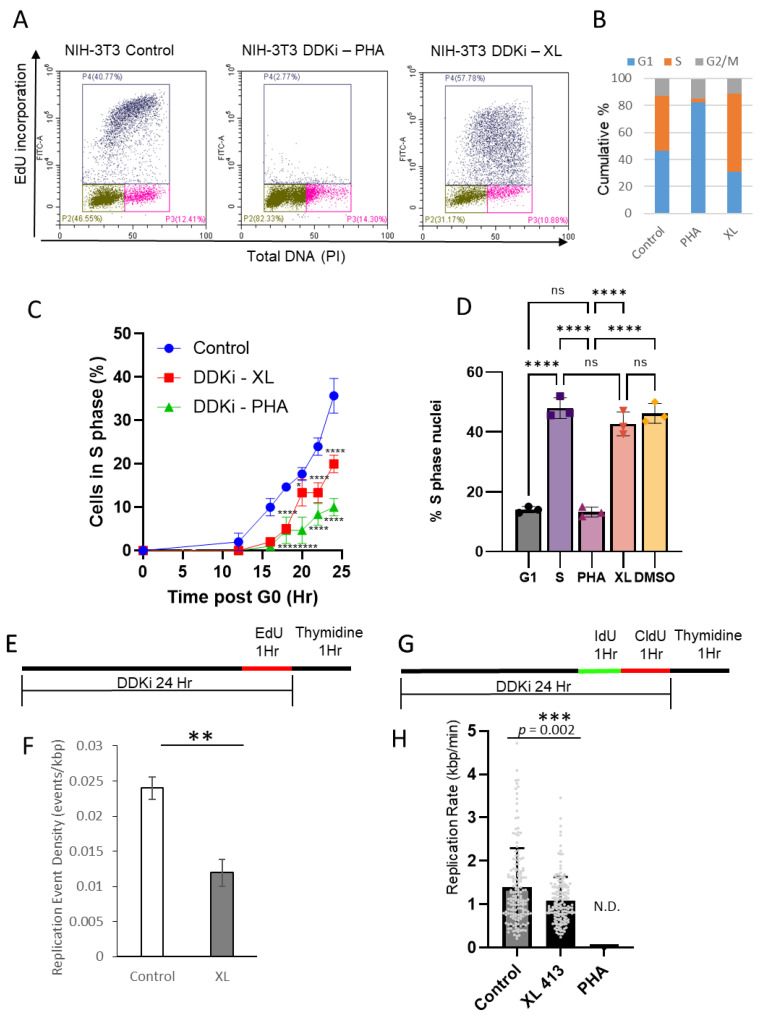
PHA-767491 and XL-413 have distinct effects on initiation and elongation phases of DNA replication. (**A**) Flow cytometry of asynchronous NIH3T3 labelled with EdU and PI to label the S-phase population. Cells were incubated with PHA-767491 or XL-413 for 24 h. The *X*-axis shows DNA content and the *Y* axis shows fluorescence intensity of EdU incorporation. (**B**) Cumulative bar charts showing G1 (blue), S-phase (Red) and G2/M (Green). (**C**) Synchronised NIH3T3 fibroblasts were released from G0 and PHA-767491 or XL-413 added 12 h after release. The percentage of S-phase cells were determined by EdU labelling at indicated time points, showing mean ± standard deviation, *n* = 3. One way ANOVA using Šídák’s multiple comparisons test where * = *p* < 0.05, **** = *p* < 0.0001. (**D**) Cell-free DNA replication assays with G1 nuclei and G1 cytosolic extracts as a baseline. G1 nuclei and S-phase extracts are used as a positive control and reactions containing 10 µM PHA-767491 or XL-413 as indicated. DMSO is used as a solvent control. The mean percentage of the population that initiates DNA replication is shown with standard deviation, where *n* = 3. Two-way ANOVA using Dunnett’s test where **** *p* < 0.0001. (**E**) Experimental overview of EdU pulse labelling and DNA combing to determine fork density. (**F**) Replication fork density is shown for control and XL413-treated cells. PHA767491 inhibited DNA replication to a level that prevented determination of replication event frequency. Data shown from 3 independent experiments, mean ± standard deviation. Significance was determined using a T-test, ** *p* < 0.01, *p* = 0.0016. (**G**) Schematic for dual labelling protocol for DNA combing for replication fork rate determination. (**H**) Bar chart showing individual events (grey dots) showing mean replication rate and standard deviation (*n* = 161 for both). No signal was detected (N.D) for PHA-767491. Statistical analysis was performed using unpaired 2-tailed *t*-test, *** *p* = 0.002 (GraphPad).

**Figure 2 biomedicines-10-02012-f002:**
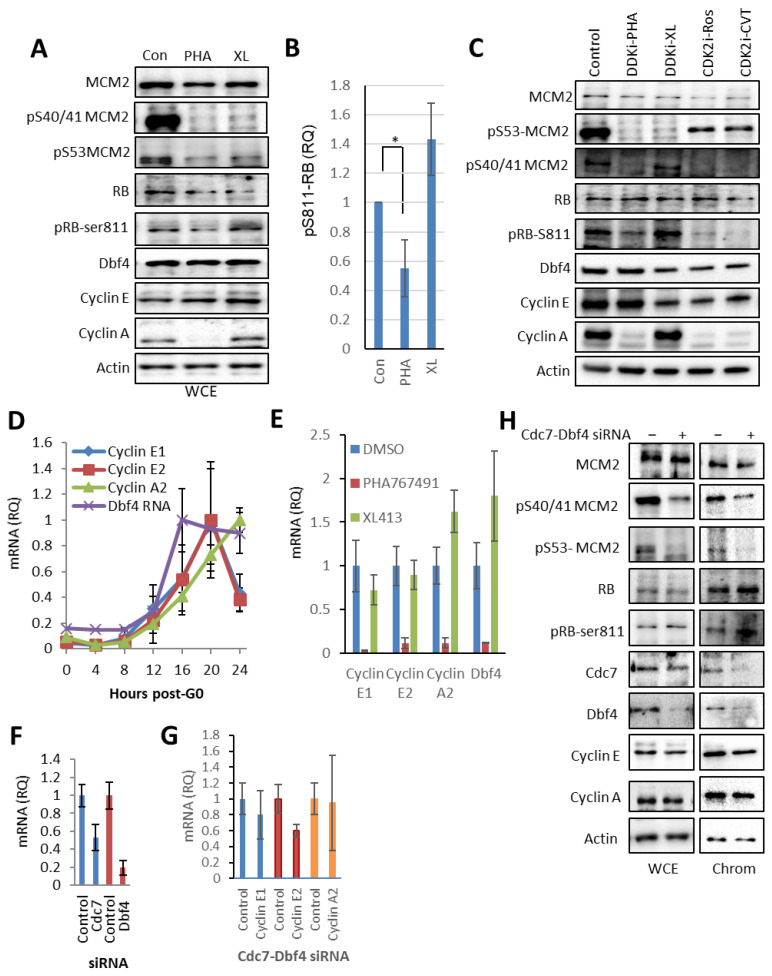
Off-target PHA-767491 activity inhibits E2F-mediated transcription of the cell cycle regulators cyclin E1, cyclin E2, Dbf4 and cyclin A2. (**A**) Immunoblot showing the levels of total MCM2, pS53-MCM2 and pS40/41-MCM2 as DDK-specific phospho-sites, cyclin E, cyclin A, Dbf4, RB, RB pS811 as CDK2 phospho-site and actin after DDK inhibitor treatment as indicated. (**B**) Quantitation of RB pS811 relative to RB total showing mean± standard deviation, *n* = 3, *p* < 0.05 (*) was determined using unpaired 2 tailed t-test. (**C**) As for A, including CDK2 inhibitors roscovitine and CVT-313. (**D**) NIH3T3 cells were synchronised in G0 and stimulated to re-enter the cell cycle. RNA samples were removed at indicated time points and transcript levels were determined by qRT-PCR. Values shown are standardised relative to maximum levels for each gene, showing mean ± standard deviation, *n* = 3. (**E**) Synchronous population was treated with DDK inhibitors 12–20 h after release from G0. The transcript levels of cyclin A2, cyclin E1, cyclin E2 and Dbf4 were monitored relative to GAPDH mRNA levels relative to the levels of the solvent control (DMSO) for PHA-767491 and XL-413 treated cells. Showing mean ± standard deviation, where *n* = 3. (**F**) 3T3 cells were synchronised and siRNA-mediated depletion of Cdc7 and Dbf4 was performed upon release from G0 phase and total RNA extracted 20 h after transfection. Transcript levels for Dbf4 and Cdc7 showing mean value ± standard deviation, which are the average of four independent experiments, each performed in triplicate. (**G**) Quantitation of E2F-regulated transcripts cyclin A2, cyclin E1 and cyclin E2 after CDC7-Dbf4 siRNA treatment from Figure 2F, showing mean ± standard deviation, *n* = 3 independent experiments of with three technical replicates. (**H**) Immunoblot of WCE and chromatin fractions of siRNA transfected cells showing Cdc7, Dbf4, MCM2 and MCM2-pSer40/41, MCM2-pSer53, cyclin E, cyclin A, Dbf4, RB, RB-pS811 and actin.

**Figure 3 biomedicines-10-02012-f003:**
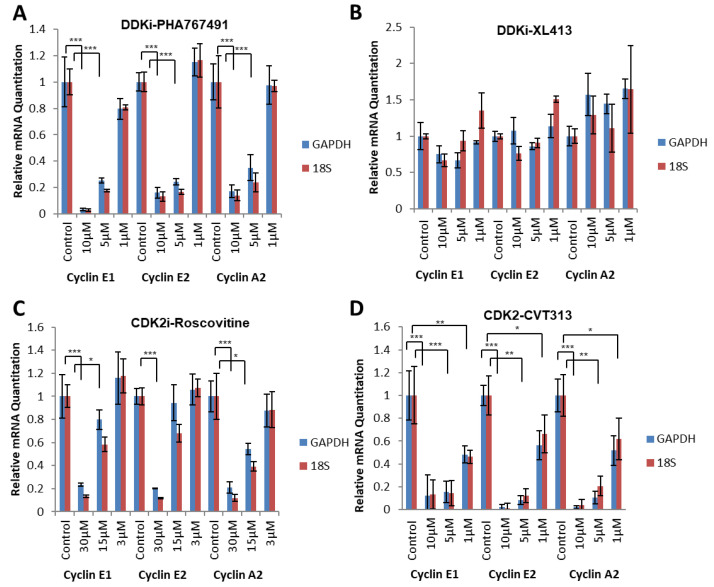
PHA-767491 transcriptional inhibition is independent from CDK9 inhibition. (**A**) qRT-PCR monitoring of cyclin E1, cyclin E2 and cyclin A2 transcription after 1, 5 and 10 µM of PHA-767491 treatment 12–20 h after G0 release. The transcript levels were compared using RNA polymerase II (GAPDH) and RNA polymerase I transcribed (18S rRNA) genes. Bars show mean value ± standard deviation, representing 3 biological replicates with 3 technical repeats in each. Statistical analysis was performed using one-way ANOVA, relative to control using Tukey’s post hoc test where * = *p* < 0.05, ** = *p* < 0.01 and *** = *p* < 0.001. (**B**) As for A using XL-413. (**C**) As for A using 3, 15 and 30 µM of Roscovitine. (**D**) As for A using CVT-313.

**Figure 4 biomedicines-10-02012-f004:**
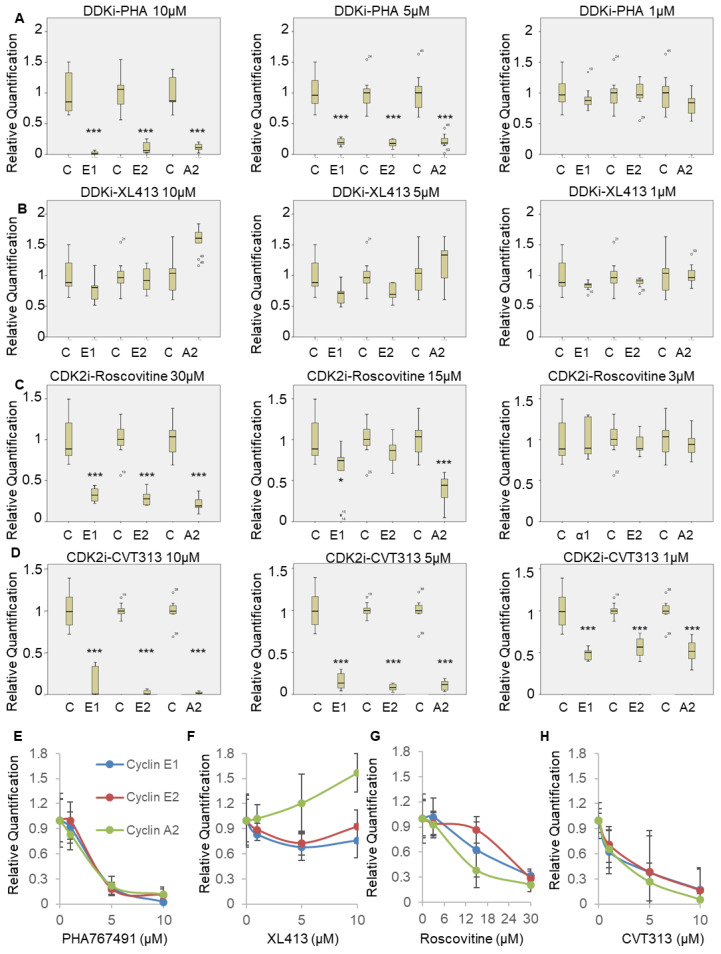
PHA-767491 inhibits E2F-mediated transcription more potently than the CDK2 inhibitors Roscovitine and CVT-313. (**A**) qRT-PCR monitoring cyclin A2, cyclin E1 and cyclin E2 in cells treated with PHA-767491 at the concentrations indicated. Transcript abundance was determined relative to GAPDH and RQ values shown. Each box plots shows median value and quartiles with standard deviation from three biological replicates, with each replicate performed in triplicate. Statistical analysis was performed using one-way ANOVA, Tukey post hoc test, shown relative to control, *p* < 0.05 is * and *p* < 0.001 is ***. (**B**) As for A except using XL-413. (**C**) As for A except cells were treated with Roscovitine. (**D**) As for A except cells were treated with CVT-313. (**E**–**H**) Dose dependence of inhibition of cyclins A2, E1 and E2 transcription for PHA-767491 (**E**), XL-413 (**F**), Roscovitine (**G**) and CVT-313 (**H**) where *n* = 3 independent replicates with bars showing mean ± standard deviation.

**Figure 5 biomedicines-10-02012-f005:**
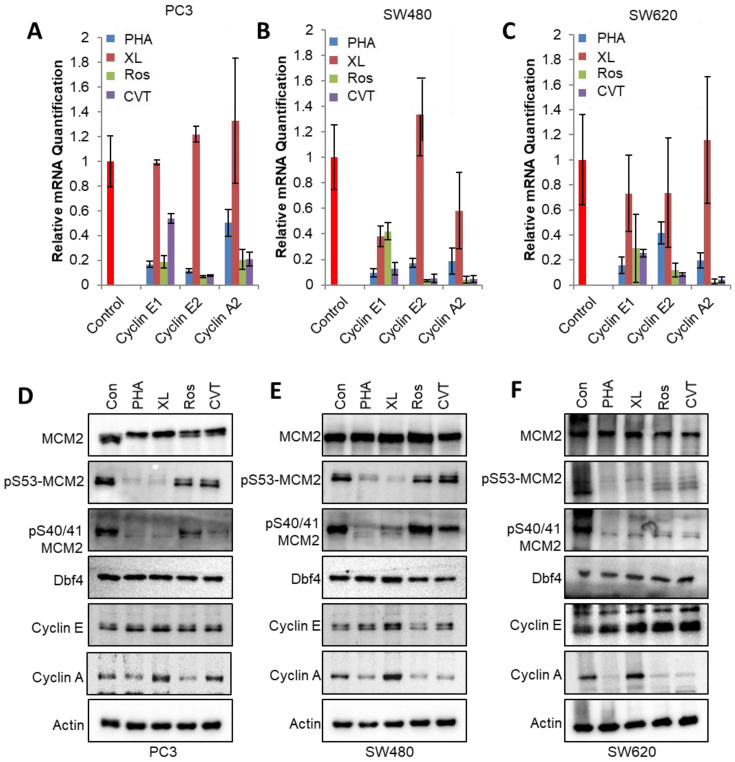
PHA-767491 reduces E2F-mediated transcription in Rb^+^ positive cancer cell lines. (**A**) Cells were synchronised in M phase using thymidine and nocodazole block, released for 4 h and treated with inhibitors for 8 h that would capture G1 transition. Prostate carcinoma cell line PC3 was treated with 10 μM XL-413, 10 μM PHA-767491, 10 μM CVT-313 or 30 μM Roscovitine as indicated. The mRNA levels of cyclins A2, E1 and E2 were monitored using qRT-PCR after 8 h of kinase inhibitor treatment. Bars show mean ± standard deviation, where *n* = 3. (**B**) As for A, using SW480 colorectal carcinoma cell line. (**C**) As for A, using colorectal carcinoma cell line SW620. (**D**) Western blot of PC3 cells treated with 10 μM XL-413, 10 μM PHA-767491, 30 μM roscovitine or 10 μM CVT-313 as indicated. The levels of DDK-mediated pMCM2-ser53, DDK and CDK-mediated phosphorylation of pMCM2-ser40/41, MCM2, cyclin E, Dbf4, cyclin A and actin are shown. (**E**) As for (**D**) using SW480 cell lines. (**F**) As for (**D**) using SW620 cell lines.

**Figure 6 biomedicines-10-02012-f006:**
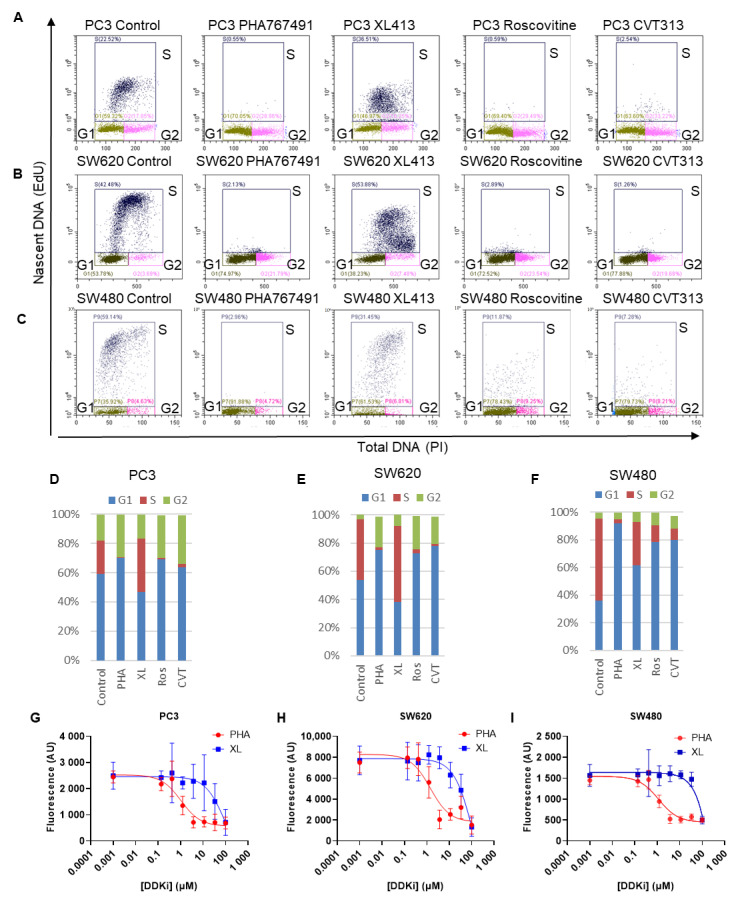
PHA-767491 potently inhibits proliferation in Rb+ prostate and colorectal cancer cell lines. (**A**) Prostate carcinoma cell line PC3 was treated with 10 μM XL-413, 10 μM PHA-767491, 10 μM CVT or 30μM CVT roscovitine or for 24 h as indicated. Cells were pulse labelled with EdU showing each cell cycle phase shown for each treatment. (**B**) As for A, except using SW620 cells. (**C**) As for A, except using SW480. (**D**) Cumulative bar charts showing percentage of G1 (blue), S phase (red) and G2/M phase (green) cells from Figure 6A. Error bars represent three biological replicas totalling nine technical repeats expressed as standard deviation. (**E**) As for (**D**), but of B. (**F**) Like D, but of C. (**G**) Dose dependence curves for IC50 calculation for PHA-767491 and XL-413 for PC3 cells, where *n* = 9 showing mean ± standard deviation. (**H**) As for H using SW480 cells. (**I**) As for H using SW620 cells.

**Figure 7 biomedicines-10-02012-f007:**
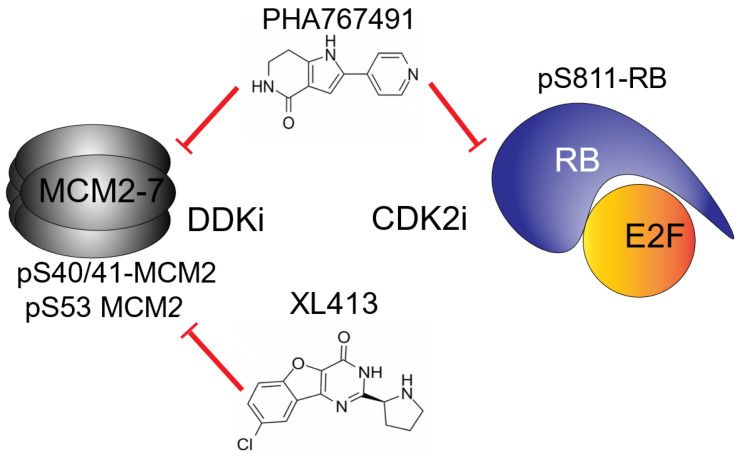
Proposed molecular mechanism for the differential responses of PHA767491 and XL413. The data presented demonstrate that both PHA-767491 and XL-413 efficiently reduce DDK-mediated phosphorylation of pS53 and pS40/41 in MCM2. This activity alone is insufficient to reduce initiation of DNA replication and cellular proliferation. PHA-767491 has the additional activity of reducing CDK2-mediated RB phosphorylation, preventing E2F-mediated transcription of G1/S cyclins. This dual activity results in enhanced efficacy and potent anti-proliferative activity in vitro.

**Table 1 biomedicines-10-02012-t001:** Small molecule DDK and CDK2 inhibitors, their targets, in vitro IC50 values and the suppliers.

Drug	Target	IC50 (Cell Free)	Provider
PHA-767491	Cdc7	10 nM	Sigma Aldrich (St. Louis, MO, USA)
XL-413	Cdc7	3.4 nM	Selleckchem (Houston, TX, USA)
Roscovitine	CDK2	700 nM	Sigma Aldrich
CVT-313	CDK2	500 nM	Selleckchem

**Table 2 biomedicines-10-02012-t002:** Antibodies used in this study. Showing target protein, supplier, order number and dilution used.

Antibody	Concentration	Provider
Cyclin E (HE12)	1:500	Abcam
Cyclin A (CY-1A)	1:500	Sigma Aldrich
Dbf4 (ab116613)	1:500	Abcam
MCM2 (BM28)	1:500	BD Transduction Lab (Franklin Lakes, NJ, USA)
pMCM2-ser53 (A300-756A)	1:500	Cell signalling (Danvers, MA, USA)
pMCM2-ser40/41 (ab70371)	1:500	Abcam
Actin (AC15)	1:2000	Sigma Aldrich
RB (ab181616)	1:500	Abcam
pRB-ser811 (ab109399)	1:500	Abcam
Cdc7 (DCS-341)	1:1000	Invitrogen
Goat Anti-Mouse IgG-HRP	1:5000	Sigma Aldrich
Goat Anti-Rabbit IgG-HRP	1:5000	Sigma Aldrich

**Table 3 biomedicines-10-02012-t003:** Antibodies used for DNA combing. Showing target protein, supplier, order number and dilution used.

Antibody	Concentration	Supplier
CldU OBT0030	1:100	BioRad (Hercules, CA, USA)
IdU (347580)	1:20	BD Biosciences (San Jose, CA, USA)
AlexaFluor 488 Chicken anti Rat (A-21470)	1:50	Life Technologies (Glasgow UK)
AlexaFluor 488 Goat anti Chicken (A-11039)	1:50	Life Technologies
AlexaFluor 633 Rabbit anti Mouse (A-21063)	1:50	Life Technologies
AlexaFluor 633 Goat anti rabbit (A-21070)	1:50	Life Technologies
AlexaFluor 568 Rabbit anti Mouse (A-11061)	1:50	Life Technologies
AlexaFluor 568 Goat anti Rabbit (A-11036)	1:50	Life Technologies
ssDNA (MAB3034)	1:100	Merck (Feltham, UK)

**Table 4 biomedicines-10-02012-t004:** IC50 values for PHA767491 and XL413 in PC3, SW480 and SW620 cell lines from Figure 6 G-I.

Cell Line	DDK Inhibitor	IC50 (µM)
SW480	PHA-767491	1.15
SW480	XL-413	>100
SW620	PHA-767491	1.4
SW620	XL-413	128
PC3	PHA-767491	1.03
PC3	XL-413	92

## Data Availability

The data presented in this study are contained within this document. If you require raw data please contact the lead author.

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
