# Peer review of "Dbf4-Cdc7 (DDK) Inhibitor PHA-767491 Displays Potent Anti-Proliferative Effects via Crosstalk with the CDK2-RB-E2F Pathway"

_biomedicines, 2022, doi:10.3390/biomedicines10082012_

Round 1
Reviewer 1 Report
The manuscript entitled: Dbf4-Cdc7 (DDK) inhibitor PHA-767491 displays potent anti-proliferative effects via crosstalk with the CDK2-Rb-E2F pathway" is exciting. The manuscript assessed the inhibitory actions of two DDK inhibitors and investigated the mechanistic details. Overall, the manuscript is well-written and scientifically sound. There are a few minor concerns. Please find my comments below.
Too many acronyms throughout the manuscript. Adding a section of abbreviations at the end will be helpful.
All in vitro/in vivo should be in italics in vitro/ in vivo.
2.1 Mention the working concentrations of pyruvate
2.5 Few crucial steps are missing in the protocol. Were the cells lysed using 1X PBS? Also, line #157, After blocking, secondary antibody? Is that correct?
Figure 1 E: Error bars are missing
Reference #21 is incomplete
Reference #37: are the page numbers correct?
Line #160: What device/Instrument was used for imaging?
Adding a hypothetical model for the mechanism of action will be advantageous.
Reviewer 2 Report
The authors showed the different inhibitory activity of two distinct DDK inhibitor chemotypes, PHA-767491 and XL-413, in cell cycle and molecular regulation. Moreover, the authors showed that PHA-767491 is a potent inhibitor of the initiation phase of DNA replication but XL413 has weak activity. In conclusion, PHA-767491 reduced E2F mediated transcription of the G1/S regulators cyclin E1, cyclin E2 and cyclin A2. However, the authors just showed that the comparison of the molecular mechanism and cell cycle regulation between these two compounds. They didn't provide the in vivo data in animal model and cancer cell death such as MTT or CCK-8 assay. Furthermore, a lot of data were unclear. For example, in Figure 2E, what is the meaning about the control in the X axis? If the control means untreated sample, why cyclin E1 was next to the control? These are different things. I am confused by your organization in many figures. Based on the basic concept in figure organization, I will reject the manuscript, because it's hard to understand it. Last, why the mRNA level of Dbf4 was decreased in PHA-treated in Figure 2E, however, the protein level of Dbf4 was not decreased in PHA-treated in Figure 2A and 2C.
Round 2
Reviewer 2 Report
I have no question.